# Exploiting Cancer’s Tactics to Make Cancer a Manageable Chronic Disease

**DOI:** 10.3390/cancers12061649

**Published:** 2020-06-22

**Authors:** Kambiz Afrasiabi, Mark E. Linskey, Yi-Hong Zhou

**Affiliations:** Brain Tumor Research Laboratory, Department of Neurological Surgery, University of California, Irvine, CA 92617, USA; afrasiabimd@gmail.com (K.A.); mlinskey@hs.uci.edu (M.E.L.)

**Keywords:** evolution of cancer therapy, surgical advances in gliomas, cancer stem cell, cancer metabolism, chromosomal instability, intra-tumoral heterogeneity, dominant cancer evolution strategies, future cancer therapeutics

## Abstract

The history of modern oncology started around eighty years ago with the introduction of cytotoxic agents such as nitrogen mustard into the clinic, followed by multi-agent chemotherapy protocols. Early success in radiation therapy in Hodgkin lymphoma gave birth to the introduction of radiation therapy into different cancer treatment protocols. Along with better understanding of cancer biology, we developed drugs targeting cancer-related cellular and genetic aberrancies. Discovery of the crucial role of vasculature in maintenance, survival, and growth of a tumor opened the way to the development of anti-angiogenic agents. A better understanding of T-cell regulatory pathways advanced immunotherapy. Awareness of stem-like cancer cells and their role in cancer metastasis and local recurrence led to the development of drugs targeting them. At the same time, sequential and rapidly accelerating advances in imaging and surgical technology have markedly increased our ability to safely remove ≥90% of tumor cells. While we have advanced our ability to kill cells from multiple directions, we have still failed to stop most types of cancer from recurring. Here we analyze the tactics employed in cancer evolution; namely, chromosomal instability (CIN), intra-tumoral heterogeneity (ITH), and cancer-specific metabolism. These tactics govern the resistance to current cancer therapeutics. It is time to focus on maximally delaying the time to recurrence, with drugs that target these fundamental tactics of cancer evolution. Understanding the control of CIN and the optimal state of ITH as the most important tactics in cancer evolution could facilitate the development of improved cancer therapeutic strategies designed to transform cancer into a manageable chronic disease.

## 1. Past and Present Cancer Therapeutics

### 1.1. Cancer Treatments Discovered over the Last 70 Years

One could historically consider the birth of the modern era of cancer therapeutics to be the serendipitous discovery of nitrogen mustard during the second world war [1]. The surprising shrinkage of lymph nodes of a lymphoma patient some 77 years ago opened a new chapter and gave birth to the modern era of cancer therapeutics [2]. Behind this, was the keen observation of scientists, following accidental explosion of mustard gas barrels on the way to Europe from America [3]. The dedication and hard work of Karnofsky and the clinical genius of Farber were the building blocks for developing the field of oncology [4,5].

The thinking at that time was that nitrogen mustard could kill the rapidly dividing malignant lymphocytes. This prompted the need for understanding the mechanism of action, dosimetry, and minimum duration of treatment [6,7]. Shortcomings and relapse started the process of thinking of drug resistance and the need to add other cytotoxic agents to treatment regimen [8]. With parallel advances in other fields such as chemistry, biochemistry, and biology, new opportunities arose [9]. This brought with it new challenges and bigger questions. Some of those questions included, why did not all the malignant cells die? How could malignant cells grow in the presence of a cytotoxic agent? Why one cytotoxic agent works in one malignant disorder, and not in another? What dose and duration would lead to the best outcome? What should we do to minimize side effects?

As knowledge regarding intracellular pathways grew, awareness of mechanism of action was taken to a higher level. Discovery of fluorinated pyrimidines during Heidelberger’s experiments on hepatoma cell lines in 1950s [10], brought 5-fluorouracil (FU) into the cancer clinical arena [11]. Indeed, as of today, 5-FU remains a major component of colon cancer treatment protocols, more than 60 years after that discovery [11]. 

The concept of multi-agent chemotherapy was born following better understanding of intracellular communication networks, as well as cell growth and division [12]. The National Cancer Institute’s initiative of exploring naturally occurring compounds in the treatment of cancer, led to the generation of an archive of more than 30,000 compounds on the shelves of NCI. A significant number of the agents currently used in the treatment of cancer come from that source [13]. Another serendipitous discovery, this time of platinum in the 1970s by Einhorn, turned the catastrophic and tragic history of testicular cancer into a unique success story [14]. Later on, and as of today, almost 50 years after that date, different platinum derivatives are the cornerstone of head and neck, lung, and gynecological cancer treatment protocols [15]. Another contribution of platinum discovery was a wake-up call that, perhaps there are other platinum-like agents for other cancers that we have not yet tried. 

Radiation therapy also came to be recognized as another cytotoxic therapy, and through time found its way into cancer treatment protocols. Its surprisingly positive effect in Hodgkin lymphoma in 1960s expedited this process [16]. Discovery of the potential synergistic effects of radiation therapy and chemotherapy, together, generated a foundation that led to the combination of chemo and radiation therapy in the treatment of head and neck, lung, and brain cancer, as well as lymphoma treatment protocols [17,18,19]

The discovery of double helix DNA by Watson and Crick in early 1950s generated the next footstep in the pathway of the evolution of cancer therapeutics thinking. Cancer was soon regarded as a disorder of genes. Inactivating mutations of tumor suppressor genes and activating mutations of oncogenes, and the resulting distortions of cellular signaling pathways gave further support to that thinking [20]. The discovery of carcinogens, specifically cigarette smoking, opened the door on research on their interaction with human genome [21]. Environmental factors, cancer prevention, screening, and early detection, followed in the footsteps of the above mentioned findings, and contribute to our present national health and cancer treatment policy guidelines.

### 1.2. Chemo- and Targeted-Therapies Co-Exist in Today’s Cancer Clinic

The next step in the evolution of cancer therapeutics was guided by the discovery of growth promoting and inhibitory pathways, cell cycle kinetics, including cyclin dependent kinases, oncogenes, tumor suppressor genes, gene regulatory machineries, and nuclear receptors. This was followed by discovery of epigenome, and micro-RNA network [22]. 

While still using chemotherapy in cancer clinic [8] and oncolytic viral vector-mediated cytotoxic therapy in clinical trials [23], we have been treating our cancer patients with a vast array of targeted therapeutic agents [24]. These agents include tyrosine kinase inhibitors [25], therapeutic antibodies [26], microRNA therapy, cytoreductive gene therapy, and gene editing for restoration or destruction of deregulated genes [27,28,29], epigenome modifiers [30], and proteosome inhibitors [31]. With an ever-increasing arsenal of cancer therapeutics and their rapid appearance in the cancer clinic, there has come significant refinement, and an explosive expansion of the number of clinical trials [32].

### 1.3. Immunotherapy Has Begun to Appear in Upfront Regimens

Publications about cancer treatment have recently started to move away from chemotherapy, and focus more on targeted therapeutics agents, and cell-based immunotherapy, tumor vaccines [33], and most recently checkpoint inhibitors as the new generation of immunotherapy in upfront regimens [34,35]. The history of immunotherapy for cancer goes back more than one hundred years. The thinking that the immune system could eliminate or prevent cancer was described in the work of some investigators who inoculated themselves with cancer cells of their patients [36]. Bacillus-Calmette-Guerrin (BCG) vaccination was also used to boost the immune system against cancer. Indeed instillation of BCG into the urinary bladder of patients diagnosed with transitional cell carcinoma of urinary bladder following surgical resection is still in use. Herpes simplex virus-based oncolytic immunotherapy is now applied in European cancer clinics for unresectable advanced stage of melanoma [37]. 

Around one in six cancers are thought to have their roots in infection. Some of the well-known examples include hepatitis B and C induced hepatocellular carcinoma, human herpesvirus-8 induced kaposi sarcoma in acquired immunodeficiency syndrome, epstein-Barr virus induced Burkitt lymphoma in sub Saharan Africa, human papillomavirus induced carcinoma of cervix and oral cavity and *Helicobacter Pylori* induced gastric maltoma. Vaccination against specific human papillomavirus serotypes has significantly reduced the incidence of squamous cell carcinoma of cervix. Because of immune dysregulation, low level of tumor antigen presentation, cross reactivity with self-antigens, and poor immune response among many other pitfalls, vaccination against the vast majority of malignancies has continued to face major challenges. High dose IL2, with or without lymphokine or anti-CD3 activated killer cells in the treatment of melanoma in 1990s led to minor response with major and life threatening toxicities. Alfa-interferon showed similar results and had similar problems. 

Immunotherapy for cancer came into focus around 30 years ago, with mononuclear cells from the peripheral blood, activated ex vivo, and then re-infused into patients with tumor. This treatment failed to achieve long-term responses [38]. Our frustration with the old generation of immunotherapy in the 1990s has most recently been replaced renewed optimism based on more recent results with checkpoint inhibitors, such as PD-1 antagonists [39]. This is the result of a more sophisticated understanding of immune regulatory pathways, since the original studies of T-cell regulation by immunologists. In essence, unleashing the immune response to tumor cells and their antigens has dramatically improved response rate and survival in a diverse group of malignancies associated with poor prognosis, including malignant melanoma [40]. Currently, there are seven approved check point inhibitors that target CTLA4, PD-1, and PD-L1 by the US Food and Drug Administration for cancer treatment ranging from non-small cell lung cancer to Merkel cell carcinoma [41]. Unfortunately, they have limited efficacy in patients with central nervous system (CNS) tumor glioblastoma or brain metastases [42]. CAR-T and BiTE are also among recent strategies in this regard [43]. These are among the most sophisticated technologies to kill cancer cells. 

The limits of immunotherapy arise from major similarities between normal cells and cancer cells, especially cancer stem cells, with little differences of their surface antigens. Cancer stem cells could easily repopulate the tumor following escape from current immunotherapeutic measures. However, the new generation of immunotherapy is a significant step in the evolution of cancer therapy, simply because we are recruiting the body’s natural defense to fight cancer. We are also trying to avoid toxicities associated with chemotherapy and radiation therapy, including generation of destructive mutations originating from the therapies themselves [44]. However, this strategy for cancer therapy has limitations and will not likely become a panacea for cancer therapy because of poor overall cancer immune responsiveness, and the relatively immune-privileged milieu of the CNS [45,46]. We have already started to face toxicities and relapse following such treatment measures.

### 1.4. Current Limitations of Anti-Angiogenesis Therapy 

The role of blood vessels in tumor progression has been investigated for more than a century [47]. Folkman’s hypothesis about the essential role of angiogenesis in solid tumor development [48] and discovery of angiogenic factor VEGF [49] initiated enthusiasm for anti-angiogenesis therapy. Bevacizumab (Avastin), a humanized anti-VEGF monoclonal antibody, is a major anti-angiogenesis drug in clinical use [50], to treat some devastating types of cancer, including non-small cell lung carcinoma, glioblastoma multiforme, ovarian cancer, metastatic colorectal cancer, metastatic breast cancer, and metastatic renal cell carcinoma. This has led to transient tumor control and palliation of clinical symptoms [51]. However, the attempts to “starve” and turn a tumor into a “dormant” disease have proven to be a failure as far as improvement of overall survival is concerned [52,53]. Once again, cancer evolves because of selection pressure favoring an emerging cellular phenotype where neoangiogenesis is not a rate-limiting issue.

Although most of the blood vessels in tumor are derived from angiogenesis, the leaky blood vessels in tumors, namely vasculogenic mimicry (VM), support rapidly growing tumors. They arise from tumor cells that have undergone endothelial trans-differentiation. With this process, neoplastic cells take on endothelial features and form abnormal blood vessels to help supply the tumor. This is an important part of tumor vascularization associated with cancer progression [54,55,56,57,58,59,60,61]. Current anti-angiogenesis agents work on suppressing normal cells to inhibit formation of blood vessels in tumor, but not on tumor cells to inhibit VM [62] or migration along the preexisting vessels of the host organ, namely vessel co-option, regarded as an alternative tumor blood supply [63]. The other reason for lack of survival improvement by anti-angiogenic therapy is failure to appreciate cancer’s capability to change, from high proliferation reliant on vascularity, to high invasion and/or avascular single cellular states triggered by the pressure of “starvation” therapy [64,65]. Starvation is a selection force in cancer evolution that pushes cancer to shift to other successful tactics to cope with the ever-changing tumor microenvironments of over-growth, hypoxia, and low pH. Anti-VM agents used to achieve vascular normalization, not only might improve anti-angiogenic therapies through synergistic and/or additive effects, when coupled with current anti-angiogenic therapies, but could significantly improve survival as well [66,67]. Recent efforts to identify anti-angiogenic compounds from natural products based on their effects on other essential tumorigenic pathways [68], and development of human fibulin-3 variant with dual antivascular function [69] beside other tumor-suppression effects in brain tumor model [70], are promising new directions in the evolution of cancer therapy.

### 1.5. Therapeutic Strategies Based on Targeting Cancer Stem Cells

Recognition of cancer as a disease of genes and dysregulated molecular pathways of cell growth guided the development of targeted therapies. Identification and characterization of a small population within neoplastic cells, described as cancer stem cells (CSC), and their role in cancer evolution and cancer recurrence [71,72,73,74] has guided the recent development of a new generation of targeted therapy against this cancer cell population. In the past decade, therapeutic agents have been developed to target CSC surface markers, crucial developmental signaling pathways for stem and progenitor cell homeostasis and function, such as the Notch, WNT, Hedgehog, and Hippo signaling cascades, as well as CSC niche and differentiation therapy. These approaches are currently at different stages of clinical development. Some have achieved promising results in certain malignancies, such as thyroid cancer and refractory desmoid tumors treated with γ-secretase inhibitors. For refractory acute myeloid leukemia (AML), agents that promote β- catenin degradation; for basal cell carcinoma, hedgehog inhibitors; and for SHH-subtype medulloblastoma and newly diagnosed AML, SMO inhibitors have been tried [75]. One has to wait for maturation of data and further developments in this field, given the high toxicity experienced by patients and potent anti-tumor activity in some preclinical models, which have failed to be reproduced so far in clinical studies.

## 2. Advances in Neurosurgical Oncology Reflect Advances in Surgical Oncology 

In contrast to the disappointing history of treating cancer patients with drugs, surgical removal of tumor remains the most effective treatment for most patients with solid cancer. Surgery is usually offered as first-line therapy for localized disease and in non-CNS locations can be curative when cancer is caught early. In the CNS in general, and other organs when cancer is not detected early enough, individual cancer cells are infiltrating into surround normal tissue and surgery for cure is no longer likely. In these circumstances, maximum tumor cytoreduction continues to have the largest intervention effect size, especially for the CNS. Below is a review of the advances in the resection of brain tumors, which reflects the advances in surgical oncology.

Over the past hundred years, glioma surgery has been greatly advanced, from essentially “exploratory” surgeries by neurosurgeons using a finger trying to discern underlying firmness or abnormal ballottability, to guided surgery by computerized tomography (CT), magnetic resonance (MR) imaging. The application of microscopic magnification and lighting visualization [76,77] and developed tool such as cavitronic ultrasonic aspirator (CUSA) [78] allow >80–90% tumor resection in proper surgical hands. 

However, this combination still required the surgeon to determine the tumor-brain interface visually and/or through tactile/haptic appreciation. Further incremental advances in this area included the development of intra-operative frameless stereotactic neuronavigation in the 1990s based on pre-operative neuro-imaging, first with proprioceptive jointed arm [79], and then with optical tracking much like an operating room optical global positioning satellite (GPS) triangulation camera tracking device [80]. Attempts to embellish this further through real-time intra-operative MR surgery has been limited by equipment, surgical tool, and microsurgical positioning and approach compromises that have hampered its overall utility [81,82]. The advent of intra-operative tumor cellular fluorescent imaging (5-ALA) has added utility for select cases [83,84]. 

Functional (Fx) MR imaging and diffusion tensor (DTI) tractography, coupled with real-time somatosensory evoked potential (SEP) motor strip cortical mapping and continuous intra-operative monitoring with SEP’s and intermittent monitoring with motor-evoked potentials (MEP) is currently the best aide to safe maximal surgical tumor cytoreduction efforts in patients under general anesthesia. Figure 1 depictures the timeline for key advances in neurosurgery.

Among the modalities utilized to treat patients with malignant gliomas (surgery, radiation therapy, chemotherapy, targeted agent therapies, immunotherapy, etc.,), surgical cytoreduction has the largest effect size. In properly selected patients based on thorough pre-operative treatment planning analysis, we can now routinely achieve a “one log kill” (90% volumetric resection) of the enhancing tumor volume. Studies have shown that significantly improved glioma outcomes begin at the 70–78% volumetric resection threshold [85,86], but increase even more at the 89–90% tumor resection level [87,88,89]. Ideally we would like to achieve a “two-log kill” (99% volumetric resection) of the enhancing tumor volume as studies show another statistical breakpoint with further improved glioma patient survival at the 98% volumetric resection level [87]. Unfortunately the jump to ≥90% volumetric resection is associated with worse early post-operative functional outcomes, that can have negative effects on survival if they do not recover [89]. While most of these functional outcomes usually normalize over 1–3 months post-operatively in many patients, so long as a vascular ischemic injury has not developed [89,90], it is an unfortunate current reality that most glioma clinical trials require a high early post-operative functional clinical status for eligibility. This means that patients that would likely benefit most from a combination of maximal surgical cytoreduction coupled with best available adjuvant therapy, do not get the latter, while those that qualify for the best available adjuvant therapy may not have had the benefit of best possible surgical cytoreduction. Even worse, given the desire to enroll as many patients in to clinical trials as possible at major cancer centers, there may be pressure to favor surgeons who maintain immediate post-operative clinical functional status through less aggressive, <90% tumor resection, leading to patients not benefiting from the intervention with the maximal size effect. This current flaw in our clinical research paradigm trying to advance the care of our patients with malignant gliomas needs to be addressed.

## 3. Exploiting Cancer’s Evolutionary Tactics for Future Cancer Therapeutics 

Our failure in curing cancer necessitates rethinking our strategy for future cancer therapeutics. If one looks at the grand scheme of the evolution of cancer therapeutics in the last almost eighty years, one comes to realize that we have gone the path of becoming more sophisticated cancer cell killers. Unfortunately, we still face seemingly insurmountable barriers, as witnessed by the fact that the vast majority of metastatic cancer patients succumb to their disease or complications of the treatment given to them [91,92]. Almost all metastatic cancers are incurable at the time of this writing, in spite of major advances in drug development and delivery [93]. This should act as the main motivation for exploring other types of approach to cancer treatment [94]. Along this path, the next reasonable step might be, turning cancer into a manageable chronic disease. 

### 3.1. Control of Chromosome Instability Rate and Modulating Intra-Tumoral Heterogeneity

No two cancer cells among billions of cancer cells comprising a tumor mass are exactly the same [95]. There is massive amount of interplay among these cells on one hand, and among the microenvironment and these cells on the other [96]. Two fundamental pillars of cancer biology have been long recognized: (1) aneuploidy and chromosomal instability (CIN) of cancer cells [97,98], and (2) intra-tumoral heterogeneity (ITH) [99,100]. 

CIN refers chromosome segregation errors in cell division, resulting in numerical and structural chromosomal abnormalities of daughter cells. Aneuploidy refers to karyotype of cells with alteration of chromosome number that is not a multiple of the haploid complement. Hence, it is different from polyploidy, which is a multiple of their haploid karyotype. Since early 50s, and with more focused studies in the recent decade, cancer has been perceived as a disease process involving chromosomes, based on the observation that most cancer cells are aneuploid [101,102]. Chromosome 8 trisomy causes adult acute myeloid leukemia [103], and some oncogenes act like carcinogen that initiates carcinogenesis by inducing aneuploidy [104]. The catalytic role of CIN in cancer development has also been suggested by a theoretical study of cancer progression [105]. This phenomenon, not only expedites cancer evolution de novo, but also re-directs its evolutionary path responding to the selective pressures of therapeutic intervention [106,107,108]. A major contributor to aneuploidy, CIN then causes a macroevolution leap, at a time scale of cell division, to cause ITH that drives cancer evolution and cancer recurrence [109,110]. These two major pillars, should serve as the target of our future cancer therapeutics. Recently, we have started to realize their contribution to our past and present cancer treatment failures [111,112,113,114]. Even though we have known about them for a long time, we have not yet developed solid therapeutic strategies addressing these two related issues. 

Mounting evidence links CIN to aggressive tumor behavior [115,116,117,118,119]. Altering survival factors through CIN as well as double minutes [120,121] and shifting cancer cellular phenotypes between the “grow” or “go” states [122,123] are all involved in the state of intra-tumoral heterogeneity [124,125]. Each is an important mechanism securing the survival and progression of cancer and recurrence over time. In glioblastoma multiforme, one of the most catastrophic cancers, the invasive stem-like tumor initiating cells (STIC) [126,127] in the tumor mass outer border and the proliferating tumor mass-forming cells (TMC) in the core of tumor mass, with their different metabolic programming and inter-changeability, could master the survival and progression game [124,128]. Cancer needs to be treated as a cellular population problem, with dynamic interplay between subpopulations, rather than just a problem of one cancer cellular phenotype. This empirical concept of tumor heterogeneity with functional tumor subpopulations playing in the evolution of cancer is supported by mathematical modeling of gene expression data of single cancer cells as well as whole tumors of large number of cases [129]. 

### 3.2. Cancer-Specific Metabolism and Energetics

Deep in the matrix of living cells, mitochondrial energetics has been linked to a wide range of diseases, including cancer [130,131]. Methodologies to monitor cancer progression have commonly exploited the altered metabolism of cancer relative to normal tissue. The preference for glycolysis over oxidative phosphorylation in neoplastic tissue is known as the Warburg effect [132,133]. The increased uptake of glucose and a structural analogue (2-fluoro-2-deoxy-D-Glucose) by cancer cell due to its altered metabolism has been exploited as a diagnostic tool, known as 18-fluorodeoxyglucose-positron emission tomography, to evaluate extent of disease and response to treatment in cancer patients [134]. 

We began to realize that exploiting cancer metabolism for clinical benefit necessitates defining the pathways that limit cancer progression and understanding the context specificity of metabolic preferences and vulnerabilities in malignant cells [135]. Antimetabolites, as the name suggests, have long been used as cancer chemotherapy by inhibiting the use of a metabolite needed for normal cellular metabolic functions [136]. With tremendous advance over the past decade in the understanding of cancer cell metabolism, we have entered a new era of targeting metabolic enzymes and their complex roles in cancer [137,138,139]. Unfortunately, current therapies targeting cancer metabolism have exhibited dismal results so far in clinical trials, despite the appealing concept and rationale for targeting metabolism [140,141,142]. 

In addition to differences in metabolism between normal and neoplastic tissues, metabolic differences exist among different tumor subpopulations that affects their interactions with tumor microenvironment [128,143]. This is reflected in the two fundamental pillars of cancer biology.

### 3.3. Cancer Therapeutic Development by Exploiting Chromosome Instability 

Cancer cells utilize gene regulatory and micro-RNA networks, which are temporally and spatially controlled during embryonic development [144,145,146]. The physical location of the cancer cell subpopulations, associated with differences in angulation, traction, and other physical parameters [147], could affect a diverse group of cellular functions, including gene regulatory mechanisms. This could potentially explain the existing differences in genetic imprinting and cellular network entropy of different cells inside the tumor mass, and consequently intra-tumoral heterogeneity [148,149]. 

Aneuploidy is a manifestation of the same fundamental change, deep in the matrix of the living cell. In cancer cells, the rate of mutational processes and chromosomal instability is affected by constant variations and increase in these changes over time, and following chemotherapy, or radiation therapy, which act as geno-toxic agents [112]. Insight into these dominant cancer evolution strategies, leading to not only cancer survival in the face of therapy, but ultimately cancer progression should guide the development of future cancer therapeutics [150]. 

CIN and aneuploidy come from errors in cell division, including mitotic checkpoint defects, aberrations in centrosome duplication cycle, altered kinetochore function, microtubule attachment defects, chromosome cohesion defects, and mutations building further genomic instability [151]. Based on partial understanding the cause of CIN, CIN-inducing drugs have been sought [152], and clinically explored, such as taxanes and other tubulin-binding drugs, aurora family kinase inhibitors and PARP inhibitors, which increase chromosome segregation errors and CIN, which ultimately lead to cell death [152,153,154,155]. Note wisely, widely used ionizing radiation in cancer treatment is among the strongest inducers of CIN [156]. CIN-inducing drugs could kill cancer cells by triggering immune response [157], which has shown synergistic effect on tumor regression with immune-checkpoint inhibition in a syngeneic mouse xenograft model of ovarian cancer [158]. However, immune evasion and ongoing CIN during immune-checkpoint inhibition might also lead to treatment failure [114]. 

Though more important in terms of achieving control of cancer plasticity and therapeutic resistance, CIN-reducing drugs are less explored. It is due to limited knowledge about how cancer, as a whole biological entity, manages to control CIN rate of heterogeneous cancer cells in an ever-changing microenvironment, in the path of cancer initiation and progression. A study suggested that replication stress causes CIN in colorectal cancer cells with silencing of CIN-suppressor genes [159]. 

Sansregret and colleagues show that one mechanism to restrain excessive CIN in tumor cells and increase fitness is through mutations in the anaphase promoting complex/cyclosome [160]. A recent study on cell cultures of glioblastoma multiforme showed that low cell-plating density caused increase of CIN rate, suggesting cancer cells’ ability to sense cue from extracellular environment to alter their CIN rate [161]. Furthermore, they found that EGF-containing fibulin-like extracellular matrix protein 1 (EFEMP1, also known as fibulin-3) played an inhibitory function of CIN triggered by low cell-plating density in vitro and low cell inoculum volume in vivo [161,162]. There remains challenges in therapeutics against CIN. Current CIN-inducing cancer therapeutic strategies is somewhat a simplistic approach, which ignores the dynamic nature of CIN. This could act as a double-edged sword. 

### 3.4. Future Cancer Therapeutics Targeting Natural Cancer Evolution Tactics

Failure of cancer therapeutics in the past and present clearly indicates that future cancer therapeutics will not win through a simple cancer-killing strategy. Future cancer therapeutics should target the natural strategies of cancer evolution, e.g., the mechanisms employed by cancer to maintain intra-tumoral heterogeneity that underlies resistance to targeted therapeutics. This would establish an equilibrium that converts the overall tumor population into an indolent and chronic neoplasm in the human body.

As shown in Figure 2, from CIN-empowered cell variables, the successful selection in favor of cancer development would simplify the tumor-ecology by streamlining subpopulation diversity down to only the essential subpopulations; to form a team of synergistically interactive functional tumor cell subpopulations that would drive the fast growth and invasive characteristics of cancer. In each stage of cancer evolution, CIN would also transiently work against itself by de-stabilizing the optimal tumor-ecology. In this scenario, selection would be directed to suppress CIN. Thus, both promotion and inhibition of CIN are important events favoring successful cancer evolution. Understanding such “Yin and Yang” reciprocal aspects of CIN could facilitate development of future cancer therapeutic strategies, which could potentially prevent cancer recurrence and progression. Supporting this theory, Zhou et al. showed extracellular control of CIN rate in glioblastoma cells, and further demonstrated that EFEMP1, which is a cell context-dependent extracellular matrix protein, functions as an inhibitor of CIN [161].

Extracellular proteins playing a dual function in cancer are particularly interesting at it relates to natural cancer evolution tactics. They could differentially modulate cancer cell populations within a tumor that are different, in karyotype and metabolism, to synergistically optimize a tumor’s opposing and balanced “grow” (cellular proliferation and neo-angiogenic dominant) or “go” (angiogenesis-independent single cell invasion and migration dominant) cellular behavior in response to changes in local microenvironment, such as deprivation of nutrients and oxygen. The cell-context dependent dual function of EFEMP1 is one such example. EFEMP1 is an important extracellular protein employed in cancer evolution via differential regulation of gene expression in different populations to achieve cell-context dependent dual function [162]. Weaponizing EFEMP1 by enhancing its tumor-suppressing role in TMC and reversing its oncogenic role in STIC of GBM is a promising, new, and novel approach to make cancer therapeutics targeting natural cancer evolution strategies [163].

The goal of future cancer therapeutics by targeting and manipulating naturally occurring cancer evolutionary tactics is not the elimination of all cancer cells. Instead, the goal is to establish a steady state between tumor population and microenvironment thus preventing the emergence of rapidly progressive disease, which often follows therapeutic interventions, such as cyto-reductive therapy, targeted therapy, and immunotherapy.

Currently targeting one population leads to escape of some cells in the same population and conversion into another in terms of functionality in tumor development. Alternatively, the resistant clones, e.g., STIC, following targeting TMC, could convert to TMC later. Consequently, there are multiple collateral and parallel escape and survival pathways available to re-form tumor mass, following current treatment strategies. The surviving cells would replenish and recapitulate the tumor mass and tumor subpopulations at a higher level of complexity and adapt to new environment [164]. If one could design a drug that targets the natural evolutionary strategies of cancer, such as cancer-specific metabolism and chromosomal instability, one could potentially slow down the process of relapse and/or progression to a significant degree.

## 4. Conclusions

We have come a long way since the serendipitous discovery of nitrogen mustard some eighty years ago. We have developed multi-agent chemotherapy protocols. We have brought radiation therapy into our cancer treatment protocols. We have developed targeted therapeutic agents, addressing a diverse category of cancer cell survival and proliferation pathways. We have tried anti-angiogenic agents, proteasome inhibitors, oncolytic viruses, as well as antibodies against inhibitors of cancer cell genetic suicide machinery. One by one, we have become more sophisticated cancer cell killers, following our natural instinct that killing the cancer cell is the answer to the problem. Unfortunately, we are facing insurmountable barriers, as our metastatic as well as locally recurrent cancer patients remain incurable.

Tumor resection remains as the most effective front-line treatment of solid cancers, which has been significantly improved in maximizing both tumor resection and patient’s safety. Recent advances in other fields, such as computer science and robotics, are opening the door on a new path in the evolution of surgical oncology. By arming our future generation surgeons with virtual reality technology and tagging the tumor cells with fluorescent antibodies, we are developing the capability of precision surgical oncology alongside precision medicine in early detection of cancer, prognosis of cancer progression, and personalization of treatment with targeted therapeutic agents and the new generation of immunotherapy. Future efforts really need to address maximizing extent of surgical resection as the intervention with the largest therapeutic effect size, without excluding patients from clinical trials as they take 1–3 months to recover functional status resulting from maximal aggressive, but ultimately safe surgery.

Advances in technology (e.g., imaging and DNA sequencing) have allowed early detection and monitoring of cancer progression. Next generation sequencing has become commonplace in community cancer clinics [165]. Knowledge of the precise molecular signature of individual cancer patient could allow tailoring and customizing treatments predicted to be most effective in combating the establishment of resistance and subsequent relapse [166]. This is one of the strongest advances in the last several years, guiding cancer specialists to refine their treatment choices of current cancer therapeutics [167]. However, without a therapeutic strategy that addresses the fundamental pillars of cancer evolution, even individualized precision medicine will likely have limited power to achieve the task. It is time to re-think the old problem of cancer-treatment from a new viewpoint, and to exploit cancer’s evolutionary and population dynamic tactics to develop better future cancer therapeutics. Perhaps we should focus on research to develop therapeutic agents that target the fundamental pillars of cancer survival and evolution, namely, CIN and ITH, even though we have known about them for several decades. Controlling CIN and ITH holds promise for solving the resistance faced by past and current cancer therapeutics. We would likely benefit more patients overall, by transforming cancer into a manageable chronic disease, rather than solely focusing on finding a complete cure “Holy Grail.”

## Figures and Tables

**Figure 1 cancers-12-01649-f001:**
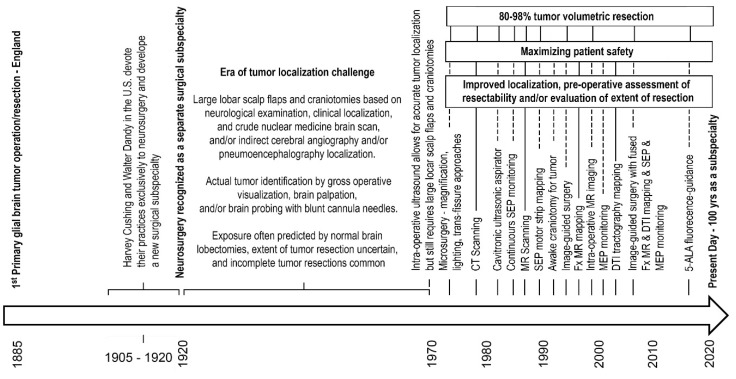
Surgical advances for gliomas timeline. Lists of abbreviations: 5-ALA, 5-aminolevulinic acid; CT, computerized tomography; DTI, diffusion tensor imaging; Fx, functional; MR, magnetic resonance; MEP, motor evoked potential; SEP, somatosensory evoked potential.

**Figure 2 cancers-12-01649-f002:**
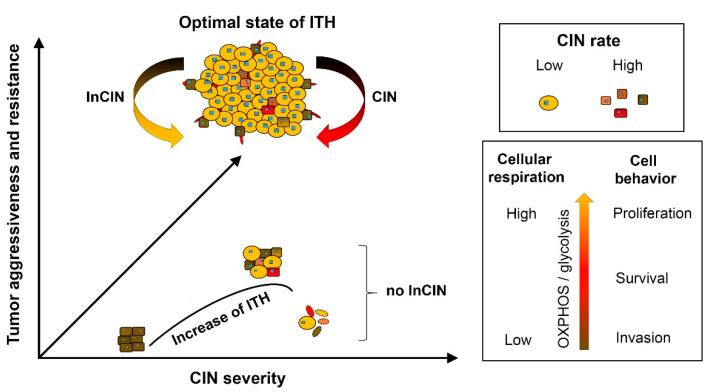
The cancer evolutionary tactic of controlling and optimizing the state of ITH. In the late steps of cancer evolution, neo-transformed cells become different functional components of a tumor, which defines intra-tumoral heterogeneity (ITH). The common metabolic feature of different tumor subpopulations is using glycolysis on top of oxidative phosphorylation (OXPHOS). The common genomic feature of tumor cells is aneuploidy. Tumor ecology streamlines tumor subpopulations to the essential ones, leading to optimal state of ITH. This leads to aggressive behavior of tumor and resistance to adverse microenvironmental factors. CIN speeds up cancer evolution by increasing ITH, while inhibition of CIN (InCIN) maintains the optimal state of ITH.

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
