# Peer review of "Exploiting Cancer’s Tactics to Make Cancer a Manageable Chronic Disease"

_cancers, 2020, doi:10.3390/cancers12061649_

Round 1

Reviewer 1 Report

The tile of this review ‘Exploiting cancer’s tactics to make cancer a chronic disease’ introduces a very interesting concept and promising therapeutic approach. However, the body of the review fails to deliver the content that one would expect based on the title and the abstract. From the abstract, the focus of the review seems to be on  ‘the tactics employed in cancer evolution; namely, chromosomal instability (CIN), intra-tumoral heterogeneity (ITH), and cancer-specific metabolism’. While the authors do a good job in summarizing current available treatments for metastatic cancers and their weaknesses, they only superficially elaborate on what should be the main body of the review (CIN, ITH and cancer-specific metabolism).

Major points:

  1. Being the focus of this review, CIN, ITH and cancer-specific metabolism should be introduced and discussed thoroughly. More details and specific examples of their impact on cancer development, progression and treatment should be given.
  2. I don’t see a place for paragraph 7 ‘Advances in neurosurgical oncology reflect advances in surgical oncology’ in this review. The focus here should be mainly on metastatic cancer which is still incurable and which the authors propose to turn into a chronic disease (lines 197-202). Surgery is usually offered as first-line therapy for localized disease and it is a curative therapy when cancer is caught early.
  3. Figure 2 is illegible (anyway I think this figure should be eliminated altogether, see point 2).

Author Response

  1. Being the focus of this review, CIN, ITH and cancer-specific metabolism should be introduced and discussed thoroughly. More details and specific examples of their impact on cancer development, progression and treatment should be given.

Response: Please see in sections 3.1, 3.2 and 3.3 in revised manuscript, we have included 19 more references on the advance of research about CIN, ITH, and cancer-specific metabolisms, listed some examples on drugs developed targeting these cancer features and clinic trails, which are detailed in these cited recent publications.

  1. I don’t see a place for paragraph 7 ‘Advances in neurosurgical oncology reflect advances in surgical oncology’ in this review. The focus here should be mainly on metastatic cancer which is still incurable and which the authors propose to turn into a chronic disease (lines 197-202). Surgery is usually offered as first-line therapy for localized disease and it is a curative therapy when cancer is caught early.

Response: Surgical oncology has been advanced greatly with proven benefit to patient on survival. We think it is important to show this advancement here in this review of cancer treatment history, while the present cancer therapeutics disappointedly failed in helping cancer patients with tumors evolved to advance stage. To increase the flow in reviewing the evolution of cancer treatment, we have moved this section (section 7 in original manuscript) as the second section, following the review of our past and present cancer therapeutics.

Importantly here to show ‘Advances in neurosurgical oncology reflect advances in surgical oncology’ in this review, as we state in conclusion “Future efforts really need to address maximizing extent of surgical resection as the intervention with the largest therapeutic effect size, without excluding patients from clinical trials as they take 1-3 months to recover functional status resulting from maximal aggressive, but ultimately safe surgery”.

Reviewer 2 Report

In the Review “Exploiting cancer’s tactics to make cancer a chronic disease “, Afrasiabi et al. attempt to address one of the most vital questions in the field of cancer research and treatment: why most of metastatic as well as locally recurrent cancer patients remain incurable? Authors acknowledge that despite all advances in technology: precision surgical oncology, precision medicine in early detection of cancer, personalization of treatment with targeted therapeutic agents and the new generation of immunotherapy, cancer remains a disease that is incurable and will remain incurable in the future but could be transformed into a manageable disease. The main conclusion of this Review is that “without a therapeutic strategy that addresses the fundamental pillars of cancer evolution, even individualized precision medicine will likely have limited power to achieve the task. It is time to re-think the old problem of cancer-treatment from a new viewpoint, and to exploit cancer’s evolutionary and population dynamic tactics to develop better future cancer therapeutics. Perhaps we should focus on research to develop therapeutic agents that target the fundamental pillars of cancer survival and evolution, namely, CIN and ITH, even though we have known about them for several decades. Controlling CIN and ITH holds promise for solving the resistance faced by past and current cancer therapeutics. We would likely benefit more patients overall, by transforming cancer into a manageable chronic disease, rather than solely focusing on finding a complete cure “Holy Grail”.

This view on the future strategy of cancer therapeutics is consistent with emerging changes of cancer paradigm, when cancer is considered not simply a disease involving abnormal cell growth with the potential to invade or spread to other parts of the body, but as a disease involving Darwinian evolution of a population of the own cells inside the body. Integration of this concept into discussion of future cancer therapeutics is the strong side of the manuscript and warrants publication of the MS.

At the same time, this MS suffers from major and minor setbacks, and can greatly benefit from clarifications and substantial changes before it can be accepted for publication in the Cancers.

  • The title of the MS, “Exploiting cancer’s tactics to make cancer a chronic disease”, is not accurate since cancer is a chronic disease. Arthritis, cardiovascular diseases, cancer and diabetes are the most frequent chronic diseases among Americans. In the text, Authors mentioned that controlling cancer evolution may open possibilities to make cancer a manageable chronic disease. “Exploiting cancer’s tactics to make cancer a manageable chronic disease” would be a more accurate title.

  • The MS consists of 7 sections and Conclusions. Sections are devoted to different types of cancer therapeutics, their history, advantages and disadvantages, future developments, etc. The main problem of this MS is that different parts of the Review are inconsistent in their quality based on the scope of reviewed publications and the depth of analysis of existing concepts, approaches, methods and treatments.

  • This inconsistency is reflected, for example, in the number of citations used in the different parts of the MS. For instance, in Section 4 (“Current limitations of anti-angiogenic therapy”) three sentences about vasculogenic mimicry (lines 156-162) are supported by nine references. At the same time, discussion in Section 6.3 of the CIN-based therapeutic strategies – the major focus of the Review – is supported by only eight relevant citations (three of which are self-citations). Unfortunately, many existing publications that address the same question and explore CIN and aneuploidy (as well as cellular adaptations to CIN and aneuploidy) as possible therapeutic targets for cancer treatment are omitted.

  • In Section 3 (“Immunotherapy has begun to appear in upfront regiments”) Authors mentioned that “currently, there are three approved check point inhibitors by the US Food and Drug Administration for cancer treatment ranging from non-small cell lung cancer to Merkel cell carcinoma” (lines 129-130). Actually, there are seven FDA approved checkpoint inhibitors that target CTLA4, PD-1, and PD-L1.

  • Section 7 (Advances in neurosurgical oncology reflect advances in surgical oncology) is a very detailed and highly proficient overview of the history of neurosurgery, discussion of the best possible strategies and the current lapses in the design of glioma clinical trials. This section, perhaps, could be published as a separate manuscript but seems to be out of the scope of this Review in its current format.

Author Response

  • The title of the MS, “Exploiting cancer’s tactics to make cancer a chronic disease”, is not accurate since cancer is a chronic disease. Arthritis, cardiovascular diseases, cancer and diabetes are the most frequent chronic diseases among Americans. In the text, Authors mentioned that controlling cancer evolution may open possibilities to make cancer a manageable chronic disease. “Exploiting cancer’s tactics to make cancer a manageable chronic disease” would be a more accurate title.Response: This is a good suggestion. We have changed the title correspondingly.
  •  
  •  
  • The MS consists of 7 sections and Conclusions. Sections are devoted to different types of cancer therapeutics, their history, advantages and disadvantages, future developments, etc. The main problem of this MS is that different parts of the Review are inconsistent in their quality based on the scope of reviewed publications and the depth of analysis of existing concepts, approaches, methods and treatments.Response: This review was prepared by two clinicians in the frontline of cancer treatment, with their expertise in medical oncology and surgical neuro-oncology, and a basic science researcher working closely with oncologists on problems of current cancer treatment. We recognize the variations in writing style and depth of reviews in different fields included in this article. To increase the flow in reviewing the evolution of cancer treatment, we have combine the past and present cancer therapeutics of sections 1-5 in original manuscript under Section 1, as five sub-sections (1.1 – 1.5), moved section 7 in original manuscript as Section 2, and followed by Section 3 with four subsections, with more in death review and discussion.
  •  
  •  
  • This inconsistency is reflected, for example, in the number of citations used in the different parts of the MS. For instance, in Section 4 (“Current limitations of anti-angiogenic therapy”) three sentences about vasculogenic mimicry (lines 156-162) are supported by nine references. At the same time, discussion in Section 6.3 of the CIN-based therapeutic strategies – the major focus of the Review – is supported by only eight relevant citations (three of which are self-citations). Unfortunately, many existing publications that address the same question and explore CIN and aneuploidy (as well as cellular adaptations to CIN and aneuploidy) as possible therapeutic targets for cancer treatment are omitted.Response: We have included 19 more recently published articles, on the advance of research about CIN, ITH, and cancer-specific metabolisms, and therapeutics tried in clinic to target CIN and cancer metabolism, as shown in Sections 3.1, 3.2. and 3.3.
  •  
  •  
  • In Section 3 (“Immunotherapy has begun to appear in upfront regiments”) Authors mentioned that “currently, there are three approved check point inhibitors by the US Food and Drug Administration for cancer treatment ranging from non-small cell lung cancer to Merkel cell carcinoma” (lines 129-130). Actually, there are seven FDA approved checkpoint inhibitors that target CTLA4, PD-1, and PD-L1.Response: Please see the correction in Section 1.3 (line 130-131) in revised manuscript.
  •  
  •  
  • Section 7 (Advances in neurosurgical oncology reflect advances in surgical oncology) is a very detailed and highly proficient overview of the history of neurosurgery, discussion of the best possible strategies and the current lapses in the design of glioma clinical trials. This section, perhaps, could be published as a separate manuscript but seems to be out of the scope of this Review in its current format.
  •  

Response: We would like to offer a complete picture for the evolution of cancer treatment, hence included the evolution of surgical oncology. As we presented in this review (line 285-297 in revised manuscript), maximal tumor resection with safety has gained benefit to cancer patients, especially patients with de novo and metastatic brain tumors. However, conflict arise between the degree of aggressive surgery, which require more recovery time, and admiration of current cancer treatment, which require immediate administration after surgery. We stated in Conclusion of revision “Future efforts really need to address maximizing extent of surgical resection as the intervention with the largest therapeutic effect size, without excluding patients from clinical trials as they take 1-3 months to recover functional status resulting from maximal aggressive, but ultimately safe surgery. “

Round 2

Reviewer 1 Report

The manuscript was improved based on the comments, with more details on CIN included. I still think that paragraph 7 (now 2), is somehow out of place. While a paragraph on surgical oncology as a curative therapy can be included, I feel that this paragraph is too long and detailed compared to the rest of the review, making it disproportionate. I suggest reducing it to a shorter summary.

Author Response

Response: Brain malignancies are the most challenging cancers for surgeons. Surgery in this organ is almost never curative. While extent of resection has the largest effect size of any intervention for CNS malignancy so far, this must be balanced against quality of life and independence of function concerns. Succeeding in with brain cancer requires and integrated multi-disciplinary approach, which includes surgery, and there is no organ system where surgical advances have been more intense and important. Failing to address surgical advances while only discussing adjuvant therapy advances would unbalance the ideal multidisciplinary approach and leave a large “hole” in the discussion. Nevertheless, we have further shortened the surgical part to comply with the reviewer’s wishes.

Reviewer 2 Report

Review “Exploiting cancer’s tactics to make cancer a manageable chronic disease “ was significantly improved. It is better organized now; raised questions and concerns were addressed.

The only part that needs additional attention is part 3 (especially, 3.1, 3.3). The MS would benefit from a thorough proof read of this part.

Part 3.1 has inaccurate statements. Some (but not all) examples: 

L317-319 "CIN refers chromosome segregation errors in cell division, resulting numerical and structural chromosomal abnormalities of daughter cells. The former is denoted as aneuploidy"  - definitions of aneuploidy and chromosomal instability are inaccurate.

L320-323 Cancer was seen as a disease involving chromosomes not only in the last decade, but since the early 50th. 

References 134 and 176 are the same.

Author Response

Response: We greatly appreciate your effort in reviewing this manuscript. Following your comments and critics, we have revised writing in sections 3.1 and 3.3. Duplicated references was a malfunction of EndNote, which has been fixed.